

# Assessment of virtual towers performed with scanning wind lidars and Ka-band radars during the XPIA experiment

Mithu Debnath[1], G. Valerio Iungo[1], W. Alan Brewer[2], Aditya Choukulkar[2], Ruben Delgado[3], Scott Gunter[4], Julie K. Lundquist[5, 6], John L. Schroeder[4], James M. Wilczak[2], and Daniel Wolfe[7]

[1]Wind Fluids and Experiments (WindFluX) Laboratory, Mechanical Engineering Department, The University of Texas at Dallas, Richardson TX
[2]National Oceanic and Atmospheric Administration, Earth Sciences Research Laboratory, Boulder CO
[3]University of Maryland Baltimore County, Baltimore MD
[4]Texas Tech University, Lubbock TX
[5]National Renewable Energy Laboratory, Golden CO
[6]Department of Atmospheric and Oceanic Sciences, University of Colorado at Boulder, Boulder CO
[7]Physical Sciences Division, National Oceanic and Atmospheric Administration, Boulder CO

*Correspondence to:* G. V. Iungo (valerio.iungo@utdallas.edu)

**Abstract.** During the eXperimental Planetary boundary layer Instrumentation Assessment (XPIA) campaign, which was carried out at the Boulder Atmospheric Observatory (BAO) in spring 2015, multiple-Doppler scanning strategies were performed with scanning wind lidars and Ka-band radars. Specifically, step-stare measurements were performed simultaneously with three scanning Doppler lidars, while two scanning Ka-band radars performed simultaneous range height indicator (RHI) scans. The XPIA experiment provided the unique opportunity to compare directly virtual tower measurements performed simultaneously with Ka-band radars and Doppler wind lidars. Furthermore, multiple-Doppler measurements were assessed against sonic anemometer data acquired from the met-tower present at the BAO site and a lidar wind profiler. This survey shows that despite the different technologies, measurement volumes and sampling periods used for the lidar and radar measurements, a great accuracy is achieved for both remote sensing techniques for probing horizontal wind speed and wind direction with the virtual tower scanning technique.

## 1   Introduction

The increasing need of monitoring the atmospheric boundary layer for a broad range of technological and scientific pursuits, such as for meteorology (Banta et al., 2002; Calhoun et al., 2006; Emeis et al., 2007; Horanyi et al., 2015; Vanderwende et al., 2015; Bonin et al., 2015), renewable energy (Thresher et al., 2008; Jones and Bouamane, 2011; Iungo et al., 2013a; Aitken et al., 2014; Iungo, 2016), air traffic management (George and Yang, 2012; Smalikho and Banakh, 2015), has led to a rapid development of remote sensing measurement techniques, such as wind lidars (Courtney et al., 2008; Cariou, 2015; Simley and Pao, 2012; Iungo and Porté-Agel, 2013b, 2014) and radars (Farnet and Stevens, 1990; O'Hora and Bech, 2007; Hirth and Schroeder, 2013; Hirth et al., 2015). Remote sensing instruments allow an easier deployment, the enhanced capability of varying deployment locations and curtailed budget as compared to classical meteorological towers.



Measurements of multiple velocity components with a single lidar or radar have been typically performed by sensing sequentially different locations of a measurement volume, upon the assumption of flow homogeneity within the measurement volume. This constraint entails limitations on the size of the measurement volume and applicability of these scanning strategies in presence of significant flow heterogeneity, such as for measurements over complex terrain (Bingöl et al., 2009) and wind
turbine wakes (Lundquist et al., 2015) .

To overcome limitations connected with multiple-component velocity measurements performed with a single instrument, multiple-Doppler scanning strategies have been explored, which require simultaneous availability of multiple instruments (Newsom et al., 2005; Mikkelsen et al., 2008; Mann et al., 2009; Carbajo-Fuertes et al., 2014; Debnath et al., 2016; Choukulkar et al., 2016). Multiple-Doppler scans consist of probing the wind velocity field at a specific location with various non-parallel
line-of-sight velocities in order to characterize the 3D nature of the atmospheric boundary layer. The number of independent non-parallel line-of-sight velocities should be equal or larger than the number of required velocity components (Newsom et al., 2008; Hill et al., 2010; Carbajo-Fuertes et al., 2014). For a specific site, at each measurement point it is possible to optimize azimuthal and elevation angles of the various line of sight directions in order to minimize error in the retrieval of the three Cartesian velocity components (Debnath et al., 2016). Accuracy in the retrieval of the three wind velocity components is
function of the norm of a matrix including trigonometric functions of elevation and azimuthal angles of the measured line of sight velocities (Debnath et al., 2016).

The virtual tower measurements presented in this paper are part of the eXperimental Planetary boundary layer Instrument Assessment (XPIA) field study, which was funded by the U.S. Department of Energy within the Atmosphere to electrons (A2e) program to estimate accuracy and capabilities of various remote sensing techniques for the characterization of complex
atmospheric flows in and near wind farms. The XPIA experiment was carried out at the National Oceanic and Atmospheric Administration (NOAA), Boulder Atmospheric Observatory (BAO) near Erie, Colorado for the period March 2 - May 31, 2015. An overview of the field campaign is provided in Lundquist et al. (2016), while a detailed analysis of several multiple-Doppler scanning strategies performed with scanning lidars was provided in Choukulkar et al. (2016) and vertical profiles of the three wind velocity components performed with triple RHI scans was presented in Debnath et al. (2016).

The XPIA experiment provided the unique opportunity of having available two Ka-band radars and three scanning wind lidars with capability of performing multiple-Doppler measurements. To the authors' knowledge, this is the first time that virtual towers performed with Ka-band radars and scanning lidars are analysed through a direct inter-comparison. Furthermore, validation of the multiple-Doppler measurements was performed against wind velocity data acquired from sonic anemometers, which were installed throughout the height of the met-tower present at the BAO site, and a lidar profiler as well.

The remainder of the paper is organized as follows: a description of the the instruments deployed for this experiment is provided in section 2. The data retrieval and assessment of the horizontal wind speed and wind direction from dual-Doppler RHI scans performed with two Ka-band radars is described in section 3, while a similar survey is then performed for the triple-Doppler step-stare scans carried out with three scanning lidars (section 4). Subsequently, inter-comparison between lidar and radar virtual tower measurements is described in section 5. Finally, concluding remarks are reported in section 6.



## 2   Experimental setup and measurement procedures

The instrumentation deployed for the XPIA experiment comprised sonic anemometers installed over the BAO met-tower, profiling lidars, radiosonde launches, microwave radiometers, and two scanning Ka-band radars. Moreover, five scanning Doppler wind lidars were deployed to explore novel scanning strategies for the characterization of ABL flows. The multiple-

Doppler measurements performed with three scanning wind lidars and two scanning Ka-band radars, which is the focus of this paper, represent one task of a broader test matrix. More details about the XPIA campaign can be found in Lundquist et al. (2016). Virtual tower measurements over the lidar supersite location (Fig. 1) were performed since UTC 19.00 March 24, 2015 until UTC 23.00 March 31, 2015. Wind data from one lidar were not available for the period March 27-28, 2015 due to a connectivity technical issue. The dataset presented in this paper is the result of a quality control process, which is function

of aerosol condition and CNR of the measured line-of-site velocities. The presented wind data are particularly valuable for assessment purpose due to the broad variability occurred both in wind speed, from 0 up to 20 m s$^{-1}$, and wind direction varying throughout the entire full circle of the wind rose.

The BAO met-tower was built in 1977 to investigate the planetary boundary layer (Kaimal and Gaynor, 1983). This 300-m tall tower has three legs spaced 3 m apart and it is instrumented with temperature and relative humidity sensors at 10 m, 100

15   m, and 300 m above ground level (AGL), while twelve 3D sonic anemometers CSAT3 by Campbell Scientific were installed at 50 m, 100 m, 150 m, 200 m, 250 m, and 300 m AGL. Six anemometers were installed on booms pointing NW (334°), which are denoted as NW sonic anemometers, while other six anemometers were installed on SE booms (154°), denoted as SE sonic anemometers. Most of the booms were 4.3 m long, while at the 250 m level the SE boom was 3.3 m long. The sonic anemometers collected data with a sampling frequency of 20 Hz, which were then tilt-corrected following the method

proposed in (Wilczak et al., 2001). The sonic anemometers were calibrated for the XPIA experiment by Campbell Scientific, with measurement resolution (maximum offset error) of 0.1 cm s$^{-1}$ (8 cm s$^{-1}$) for the horizontal wind speed and 0.05 cm s$^{-1}$ (4 cm s$^{-1}$) for the vertical velocity (McCaffrey et al., 2016).

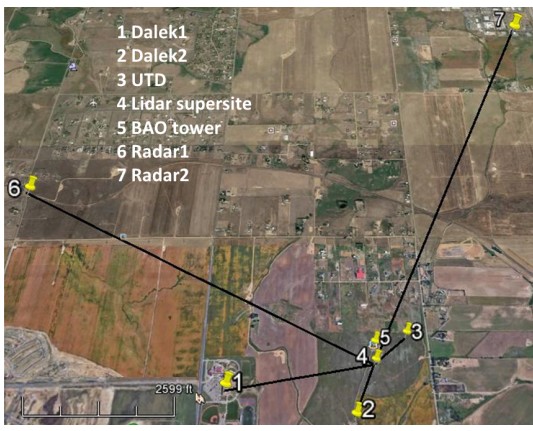

**Figure 1.** Map of the setup for the virtual tower measurements performed over the lidar supersite location during the XPIA experiment.



Vertical profiles of the three velocity components were performed with one Leosphere WINDCUBE Offshore 8.66 profiling lidar, which is denoted as V2 lidar. Wind velocity measurements were carried out with the Doppler beam swinging (DBS) technique (Courtney et al., 2008; Rao et al., 2008) with an elevation angle from vertical of 28°, sampling frequency of 1 Hz and with the range gates centered at 11 vertical heights (40 m, 50 m, 60 m, 80 m, 100 m, 120 m, 140 m, 150 m, 160 m, 180 m, 200 m). The profiler lidar was deployed at the location referred to as lidar supersite (Fig. 1), whose GPS coordinates are reported in Table 1.

Two Texas Tech University Ka-band mobile Doppler radars (Hirth and Schroeder, 2013; Hirth et al., 2015; Gunter et al., 2015) were deployed during XPIA. During the experiment, the pulse repetition frequency (15 kHz), pulse width (20 ms) and range resolution (15 m) were maintained. Radar1 (Radar2) was deployed 3.192 km (3.9 km) northwest (north) of the BAO tower (Fig. 1). Considering the 0.33° half-power beam width of the radars, these distances yielded an azimuthal resolution of 18 m (22 m) for Radar1 (Radar2) at the BAO tower location. Simultaneous RHI scans were performed by focusing both radars over the lidar supersite location by setting the radar azimuthal angles reported in Table 2, sampling rate equal to 5 Hz and sampling period of 3.3 s. After quality control analysis of the radar measurements, wind data from the two Ka-band radars for the period 25 March 1320 UTC to 1507 UTC and for heights ranging from 10 m to 490 m height with 20 m interval were available for this particular study.

**Table 1.** GPS locations of the three scanning Doppler wind lidars, wind lidar profiler, Ka-band radars and BAO tower.

|  | Longitude | Latitude | Elevation |
|---|---|---|---|
| UTD | W 105°0′3.99″ | N 40°3′2.32″ | 1578 m |
| Dalek1 | W 105°0′55.64″ | N 40°2′51.75″ | 1578 m |
| Dalek2 | W 105°0′20.65″ | N 40°2′43.09″ | 1585 m |
| BAO tower | W 105°0′13.82″ | N 40°2′0.13″ | 1579 m |
| Ka-band radar 1 | W 105°2′13.85″ | N 40°3′43.70″ | 1548 m |
| Ka-band radar 2 | W 104°59′2.98″ | N 40°4′49.51″ | 1538 m |
| Lidar supersite | W 105°0′14.36″ | N 40°2′55.72″ | 1580 m |

**Table 2.** Distance and azimuthal angle of the various instruments from the virtual tower location referred to as lidar supersite.

|  | Distance (m) | Azimuth angle (°) |
|---|---|---|
| UTD | 322 | 234.93 |
| Dalek1 | 985 | 85.95 |
| Dalek2 | 422 | 29.7 |
| Radar1 | 3192 | 118 |
| Radar2 | 3900 | 204 |
| BAO tower | 134 | 181 |





Three scanning Doppler wind lidars Leosphere WINDCUBE 200S (University of Texas at Dallas (UTD), NOAA Dalek1, NOAA Dalek2) were deployed for this experiment. Wind measurements were performed by means of eye-safe laser with a pulse energy of 0.1 mJ and wavelength of 1.54 $\mu$m. Measurements were acquired by using an accumulation time of 0.5 s and gate length of 50 m. Locations of the three scanning Doppler wind lidars are shown in Fig. 1, while their GPS positions are
reported in Table 1 and azimuthal angles with respect to the virtual tower location in Table 2. Accuracy in the radial velocity of each scanning lidar is always higher than 0.3 m s$^{-1}$ for -25 dB CNR threshold for the line-of-sight velocity Choukulkar et al. (2016). Squareness, precision and repeatability tests indicate an absolute pointing accuracy of about 0.15°. All the scanning lidars performed fixed point measurements at different heights over the lidar supersite location (Fig. 1) during the time period 0000-2400 UTC on March 25, 2015. Lidar measurements were performed at six different heights from 100 m to 200 m with
20 m steps.

The collected lidar data were further post-processed only if the carrier-to-noise ratio of the lidar signal was larger than -25 dB (Carbajo-Fuertes et al., 2014). The post-processing from the three radial velocities, $U_r$, to Cartesian wind velocity components, $U, V, W$, was carried out following the standard triple-Doppler retrieval (Mikkelsen et al., 2008; Mann et al., 2009; Carbajo-Fuertes et al., 2014; Debnath et al., 2016; Choukulkar et al., 2016; Simley et al., 2016) by means of the following equations:

$$
\begin{bmatrix} U \\ V \\ W \end{bmatrix} = \begin{bmatrix} cos(\phi_{UTD}) * cos(\theta_{UTD}) & cos(\phi_{UTD}) * sin(\theta_{UTD}) & sin(\phi_{UTD}) \\ cos(\phi_{D1}) * cos(\theta_{D1}) & cos(\phi_{D1}) * sin(\theta_{D1}) & sin(\phi_{D1}) \\ cos(\phi_{D2}) * cos(\theta_{D2}) & cos(\phi_{D2}) * sin(\theta_{D2}) & sin(\phi_{D2}) \end{bmatrix}^{-1} \times \begin{bmatrix} U_r^{UTD} \\ U_r^{D1} \\ U_r^{D2} \end{bmatrix} \tag{1}
$$

where $\phi$, $\theta$ represent elevation and azimuthal angles, respectively, of the various lidars indicated as suffix. The accuracy in the retrieval of the three velocity components was estimated for the different heights through the $L_2$-norm of the rows of the matrix in Eq. 1 including trigonometric functions of $\phi$ and $\theta$ of the various lidars. Error in the velocity retrieval increases for
values diverging from 1. In Table 3, it is shown that for this setup, the accuracy in the retrieval of the horizontal wind speed components is roughly unchanged for the different heights, while accuracy is improved with increasing heights for the vertical velocity, which is consequent to the higher elevation angles of the three lidars.

**Table 3.** Error analysis on the retrieval of the wind velocity components from triple-Doppler lidar measurements for different heights consequent to azimuthal and elevation angles of the three lidars.

| Height (m) | $U$ | $V$ | $W$ |
| --- | --- | --- | --- |
| 100 | 0.85 | 1.95 | 4.80 |
| 120 | 0.85 | 1.97 | 4.05 |
| 140 | 0.86 | 2.00 | 3.52 |
| 160 | 0.86 | 2.01 | 3.13 |
| 180 | 0.87 | 2.03 | 2.83 |
| 200 | 0.87 | 2.06 | 2.59 |





For each height of the virtual tower and each lidar, the closest range gate to the considered measurement point is selected for the data retrieval. The maximum horizontal distance of a gate centroid from the respective tower measurement point is 19 m, while the vertical one is always smaller than 10 m. The sampling period at each measurement point was 25 s, while the total time required to perform one virtual tower was on average 127 s. The three scanning lidars were not synchronized, thus the overlapping time at each measurement location was generally smaller than the prefixed period of 25 s. Histogram of the overlapping time is reported in Fig. 2, which shows a mean value of 16.3 s and a standard deviation of 3.6 s.

Bias errors in laser pointing and in the line-of-sight velocity, which were evaluated through preliminary tests (Lundquist et al., 2016) and reported in Table 4, were considered for the data retrieval. A bias in the radial velocity of the UTD lidar was due to improper calibration of the frequency chirp in the laser pulse, which was stable and reproducible in several tests, and could simply be subtracted out of the lidar measurements.

## 3 Assessment of radar virtual tower measurements

Assessment of the lidar and radar virtual towers is performed against the wind velocity components acquired through the sonic anemometers deployed throughout the height of the BAO met-tower and vertical profiles of the 3D wind velocity sampled with a lidar profiler deployed at the lidar supersite location. These data acquired are shown in Fig. 3 for the height of 150 m. In this figure, ranges of the wind direction for which the sonic anemometers may experience tower wake effects are reported with shaded areas Lundquist2016, McCaffrey2016b. A general good agreement is observed among the different instruments for both horizontal wind speed, $U_h$, and wind direction for the entire duration of the experiment.

**Table 4.** Bias errors used for the triple Doppler data retrieval.

|  | Scanner height (m) | Azimuth (°) | Elevation (°) | los velocity (m s$^{-1}$) |
|---|---|---|---|---|
| UTD | 1.37 | 4.93 | -0.89 | 0.6 |
| Dalek1 | 1.37 | 3.45 | 0.0 | 0.0 |
| Dalek2 | 1.37 | 7.70 | 0.0 | 0.0 |

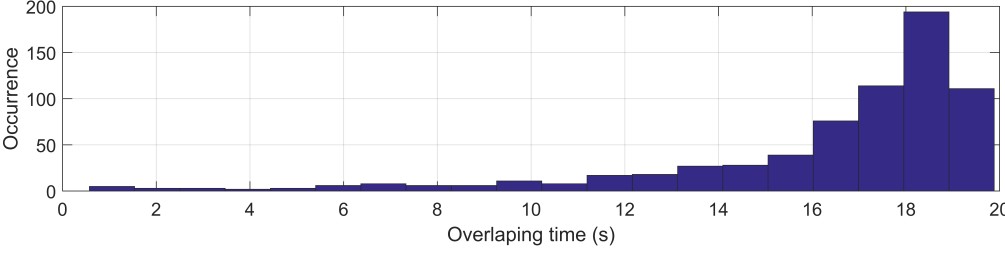

**Figure 2.** Histogram of the overlapping time of the step-stare measurements among the three lidars.



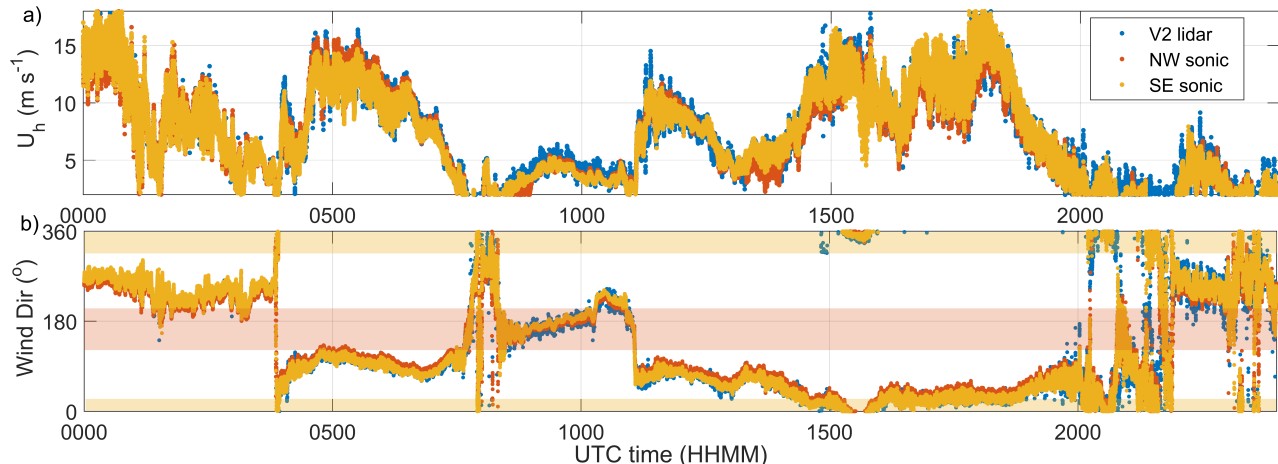

**Figure 3.** Wind velocity data acquired from sonic anemometers and lidar profiler at 150 m height: a) horizontal wind speed $U_h (m\ s^{-1})$; b) wind direction ($^{o}$). The red and yellow shaded areas highlight wind directions for which NW and SE sonic anemometers, respectively, may experience tower wake effects.

**Table 5.** Linear regression analysis among the V2 lidar, SE and NW sonic anemometers data for 24-hour data reported in Fig. 3.

| Height (m) | $U_h$ R$^2$(slope) | wind Dir R$^2$(slope) |
|---|---|---|
| | SE sonic vs. NW sonic | |
| 100 | 0.97 (1.00 ) | 0.97 (1.06) |
| 150 | 0.97 ( 1.00) | 0.97 (1.07) |
| 200 | 0.98 (1.00 ) | 0.96 (1.08) |
| All heights together | 0.97 ( 1.00) | 0.97 (1.07) |
| | V2 lidar vs. NW sonic | |
| 100 | 0.92 (0.94) | 0.88 (0.96) |
| 150 | 0.91 (0.93) | 0.92 (1.00) |
| 200 | 0.86 (0.80) | 0.94 (0.96) |
| All heights together | 0.90 (0.90) | 0.91 (0.97) |
| | V2 lidar vs. SE sonic | |
| 100 | 0.92 (0.93) | 0.89 (0.90) |
| 150 | 0.90 (0.93) | 0.93 (0.96) |
| 200 | 0.76 (0.77) | 0.96 (0.97) |
| All heights together | 0.90 (0.91) | 0.91 (0.92) |

In order to perform comparison and linear regression analysis between wind data acquired from different instruments, data acquired from instruments with a higher sampling frequency are averaged over the corresponding sampling period of





instruments with a lower sampling frequency. For instance, sonic anemometer data acquired with a sampling frequency of 20 Hz are averaged over periods with duration of 1 s for comparison with V2 lidar data acquired with a sampling frequency of 1 Hz.

Linear regression analysis performed between sonic anemometer and the V2 lidar data generally shows a good correlation

among the different instruments for the different heights. In Table 5, the slope and $R^2$ values resulting from the linear regression analysis are reported for the different heights and as overall ensemble statistics. This analysis shows that the confidence level for the assessment survey of the virtual tower measurements performed with lidars and radars against sonic anemometer and lidar profiler data is adequate.

In this section, we present the assessment of the dual-Doppler measurements performed with the two Ka-band radars against

sonic anemometer and lidar profiler wind velocity data. For the retrieval of the horizontal wind speed and wind direction through the dual-Doppler technique, the vertical velocity is assumed to be negligible, which allows dropping the last row and one column in Eq. 1. In Fig. 4, horizontal wind speed and wind direction at 150 m height retrieved from the above-mentioned instruments are compared. A qualitative general good agreement can already be perceived. In order to achieve a more quantitative characterization of the accuracy in the dual-Doppler retrieval performed on the radar data, a linear regression

analysis was then performed for both horizontal wind speed and wind direction. In order to compare radar data with sonic and V2 lidar data over different heights, a 1D linear extrapolation was performed for each timestamp in order to estimate radar wind data for the heights probed by the sonic anemometers and the V2 lidar. The correlation between the radar data and the other reference instruments is generally very high, as shown in Fig. 5, with a correlation always larger than 91%. Slope and $R^2$ values resulting from the linear regression analysis among dual-Doppler radar data, sonic anemometer and V2 lidar data

are then reported in Table 6 for the various heights and as ensemble statistics. Again, a good agreement between radar and

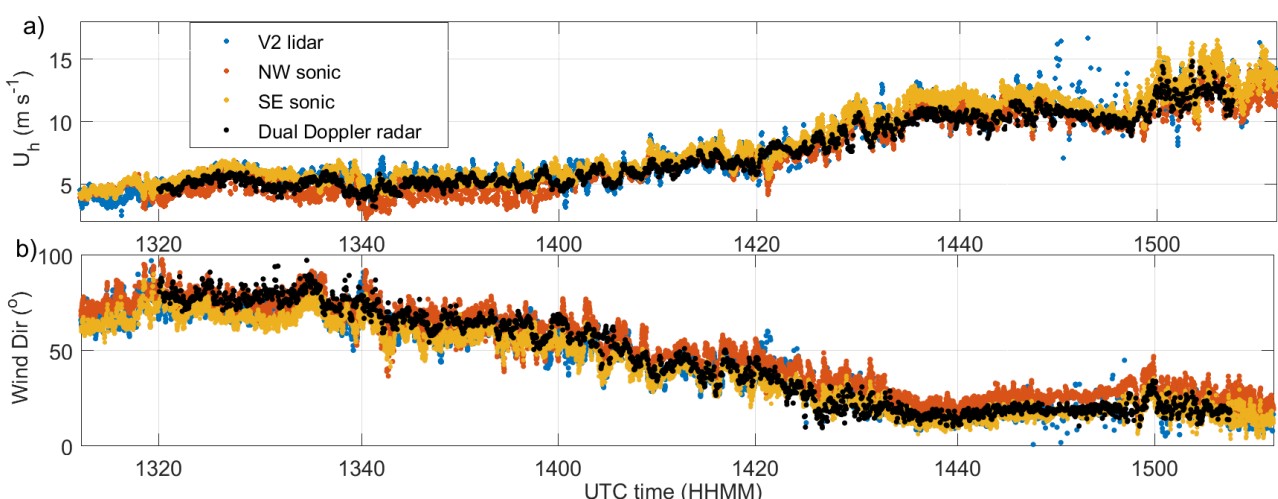

**Figure 4.** Dual Doppler radar measurements at 150 m height compared with sonic anemometer and V2 lidar data: a) Horizontal wind speed $U_h(m\ s^{-1})$; b) wind direction ($^o$).





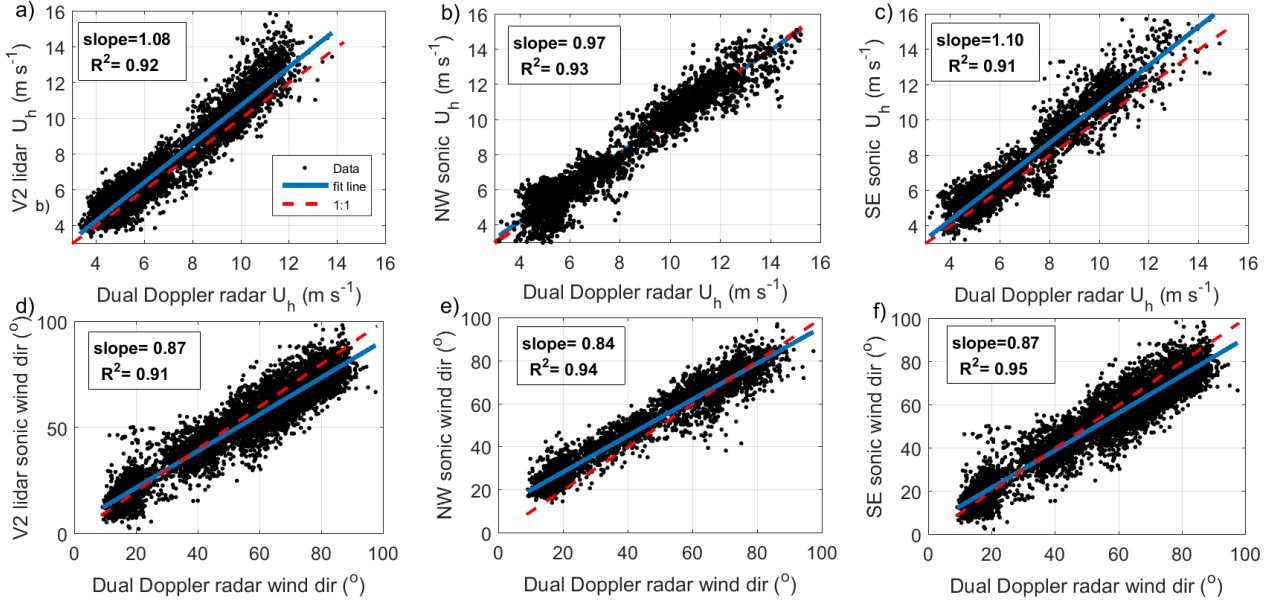

**Figure 5.** Linear regression analysis of the dual-Doppler radar retrieval against sonic anemometer and V2 lidar data for all the tested heights: a), b), c) horizontal wind speed $U_h (m\ s^{-1})$; d), e), f) wind direction $(^o)$.

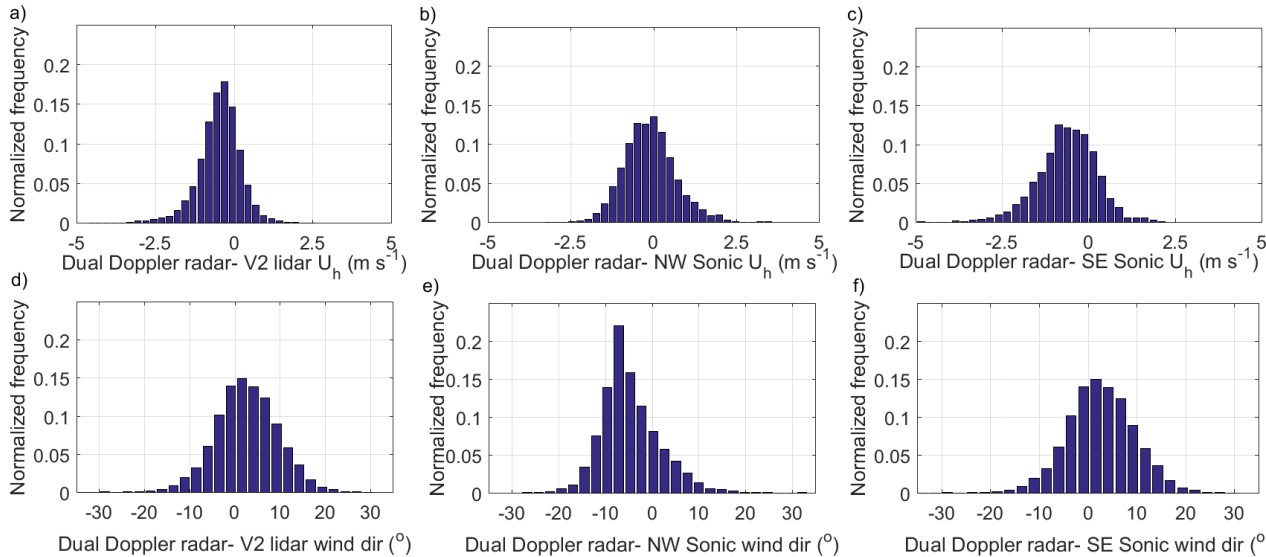

**Figure 6.** Difference of dual-Doppler radar retrieval with the reference instruments i.e sonic anemometers and V2 lidar for all the tested heights. a), b), c) Horizontal wind speed $U_h (m\ s^{-1})$; d), e), f) wind direction $(^o)$.



**Table 6.** Linear regression analysis of Ka-band radars against sonic anemometer and V2 lidar data.

| Height (m) | $U_h$ $R^2$(slope) | wind Dir $R^2$(slope) |
|---|---|---|
| Dual-Doppler radar vs. V2 lidar | | |
| 100 | 0.89 (1.02) | 0.90 (0.90) |
| 120 | 0.93 (1.04) | 0.92 (0.90) |
| 140 | 0.93 (1.06) | 0.91 (0.86) |
| 150 | 0.92 (1.06) | 0.92 (0.89) |
| 160 | 0.93 (1.08) | 0.92 (0.86) |
| 180 | 0.93 (1.12) | 0.91 (0.84) |
| 200 | 0.93 (1.12) | 0.9 (0.82) |
| All heights together | 0.92 (1.08) | 0.91 (0.87) |
| Dual-Doppler radar vs. NW sonic | | |
| 100 | - (-) | - (-) |
| 150 | 0.91 (1.03) | 0.93 (0.83) |
| 200 | 0.95 (0.92) | 0.96 (0.87) |
| All heights together | 0.93 (0.97) | 0.94 (0.84) |
| Dual-Doppler radar vs. SE sonic | | |
| 100 | 0.87 (1.10) | 0.95 (0.89) |
| 150 | 0.93 (1.07) | 0.93 (0.86) |
| 200 | - (-) | - (-) |
| All heights together | 0.91 (1.10) | 0.95 (0.87) |

reference instrument data is generally achieved throughout the height of the virtual tower and without any noticeable trend in the vertical direction.

Finally, histograms of the difference between the wind horizontal wind speed and direction measured through the dual-Doppler radar measurements and the reference instruments are reported in Fig. 6. For the horizontal wind speed the mean difference is of -0.47 m s$^{-1}$, -0.11 m s$^{-1}$, -0.63 m s$^{-1}$ compared with V2 lidar, NW sonic and SE sonic, respectively, with standard deviation of 0.68 m s$^{-1}$, 0.78 m s$^{-1}$ and 0.86 m s$^{-1}$. A similar analysis for the wind direction leads to a mean difference of 2.6$^o$, -4.75$^o$, 2.60$^o$ compared with V2 lidar, NW sonic and SE sonic, respectively, with standard deviation of 6.98$^o$, 6.28$^o$, 6.98$^o$.

## 4 Retrieval and assessment of triple-Doppler lidar measurements

In this section, we present an assessment study of the triple-Doppler lidar measurements, which were performed with three scanning Doppler lidars to retrieve the three velocity components. As for the previous section, assessment of triple-Doppler data is carried out against sonic anemometer and lidar profiler data. In Fig. 7a, the light-of-sight velocities are reported for





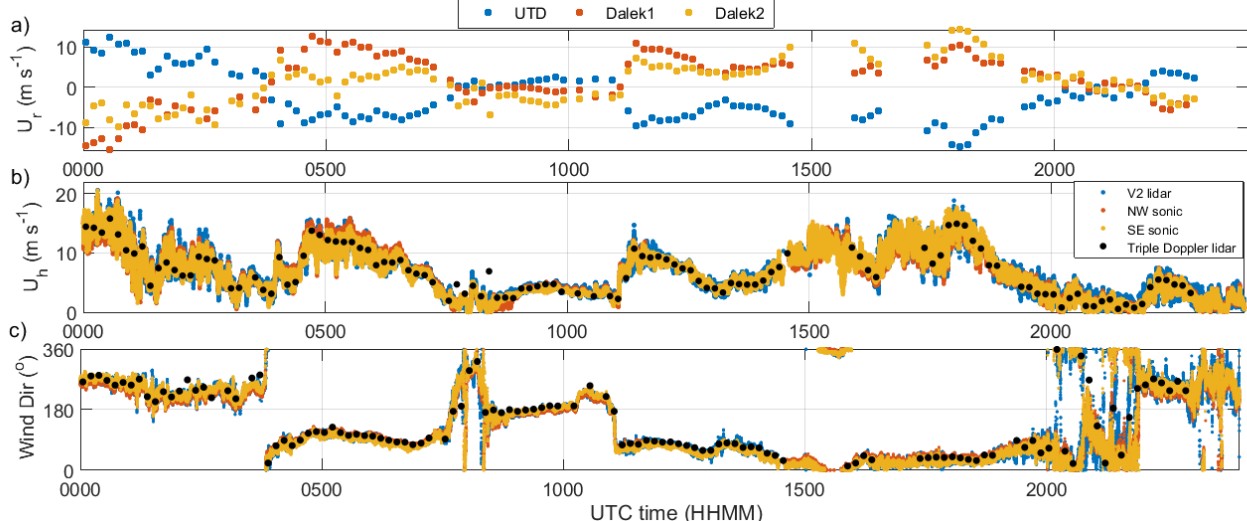

**Figure 7.** Triple-Doppler lidar measurements at 100 m height and assessment against sonic anemometer and lidar profiler data: a) line-of-sight velocities from the three scanning lidars; b) horizontal wind speed $U_h(m\ s^{-1})$; b) wind direction ($^o$).

the measurements carried out at 100-m height during the entire period of the experiments. The wind data considered for the triple-Doppler retrieval are first quality controlled as a function of the carrier-to-noise ratio (minimum value of -25 dB), then averaged over the actual sampling period, which is defined as the time for which the three lidars measured simultaneously over the location of interest. Statistics of the actual sampling period, i.e. of the overlapping time among the three scanning lidars, have been already presented in Fig. 2.

The retrieved vertical velocity was assessed only against the V2 lidar data, because the horizontal distance of 134 m between the BAO tower and the lidar supersite location (see Table 2) as well as the different averaging volume of each instrument lead to poorer agreement between sonic anemometer and triple-Doppler lidar data, as reported in Table 7. The linear regression in the vertical velocity with the V2 lidar data, in contrast, shows a good agreement for the height of 200 m with a slope of 0.94 and a correlation of $R^2$=0.79. As prefigured through the error analysis presented in Table 3, the reduced elevation angles of the lidar laser beams for smaller heights lead to a rapid decay in the accuracy for the retrieval of the vertical velocity through the triple-Doppler lidar measurements.

The horizontal wind speed and direction retrieved through the triple-Doppler lidar measurements are reported in Figs. 7b and c, respectively. In these figures, the respective velocity data directly measured at 100 m height highlight that, just as for more traditional instruments, such as sonic anemometers and the lidar profiler, the triple-Doppler measurement technique allows characterization of a significant daily variability in wind velocity from quiescent conditions up to about 20 m s$^{-1}$. Good performance is also observed for the characterization of the wind direction. Indeed, during the experiment, wind direction varied all around the full angle of the wind rose, and the triple-Doppler measurements were able to detect crisply the different angles of the wind direction and following its variability as a function of time.





**Table 7.** Linear regression analysis of triple-Doppler lidar data against the reference instruments, namely sonic anemometers and V2 lidar.

| Height (m) | $U_h$ $R^2$(slope) | Wind Dir. $R^2$(slope) | W $R^2$(slope) |
|---|---|---|---|
| | Triple-Doppler lidar vs. V2 lidar | | |
| 100 | 0.94 (0.99) | 0.92 (0.97) | 0.01 (0.13) |
| 120 | 0.97 (0.99) | 0.93 (0.95) | 0.27 (0.32) |
| 140 | 0.97 (0.97) | 0.85 (0.97) | 0.57 (0.52) |
| 160 | 0.94 (0.97) | 0.88 (0.93) | 0.62 (0.63) |
| 180 | 0.95 (1.00) | 0.95 (0.92) | 0.77 (0.68) |
| 200 | 0.93 (1.07) | 0.99 (1.00) | 0.79 (0.94) |
| All heights together | 0.96 (0.95) | 0.90 (0.97) | 0.49 (0.42) |
| | Triple-Doppler lidar vs. NW sonic | | |
| 100 | 0.92 (0.89) | 0.85 (0.90) | 0.008 (0.04) |
| 200 | 0.90 (1.12) | 0.90 (0.91) | 0.13 (0.12) |
| All heights together | 0.89 (1.02) | 0.87 (0.90) | 0.09 (0.1) |
| | Triple-Doppler lidar vs. SE sonic | | |
| 100 | 0.89 (1.12) | 0.84 (0.91) | 0.005 (0.012) |
| 200 | 0.9 (0.94) | 0.93 (1.00) | 0.092 (0.11)) |
| All heights together | 0.89 (1.01) | 0.87 (0.95) | 0.03 (0.08) |

Accuracy in the triple-Doppler retrieval of wind horizontal wind speed and direction is then quantitatively characterized through a linear regression analysis, which was performed for all the heights under examination against sonic anemometer and lidar profiler data (Fig. 8). Starting with a comparison with the V2 lidar profiler data located over the lidar supersite location, a very good agreement is estimated between these measurement techniques. For the horizontal wind speed, the slope is 0.96 with correlation of $R^2$=0.95, while for the wind direction the slope is 0.97 and correlation of $R^2$=0.9.

Moving to the linear regression of the triple-Doppler lidar against sonic anemometer data (Fig. 8), the horizontal distance of 134 m between the BAO tower and the lidar supersite location, where all the scanning lidars are focused, does not affect significantly agreement between measurements obtained from the various instruments. Indeed, the slope for the wind velocity varies between 0.9 and 1.02, with correlation always larger than $R^2$=0.89. For the wind direction, the slope is 0.9 and 0.95 for the linear regression against NW ans SE sonic anemometers, respectively, while the correlation is $R^2$=0.87.

Results of the linear regression analysis for the measurements carried out at different heights are reported in Table 7. Considering the data against the V2 lidar, slope for the horizontal wind speed is always very close to one, with minimum value of 0.97 and maximum value of 1.07, while the correlation is always larger than $R^2$=0.93. For the wind direction, a reduced level of accuracy is estimated with correlation larger than $R^2$=0.88 but with slope still very close to 1. As for the error analysis consequent to the setup of the three lidars (see Table 3), accuracy in the measurements for both horizontal wind speed and wind direction is not noticeably changed for the locations at different heights.




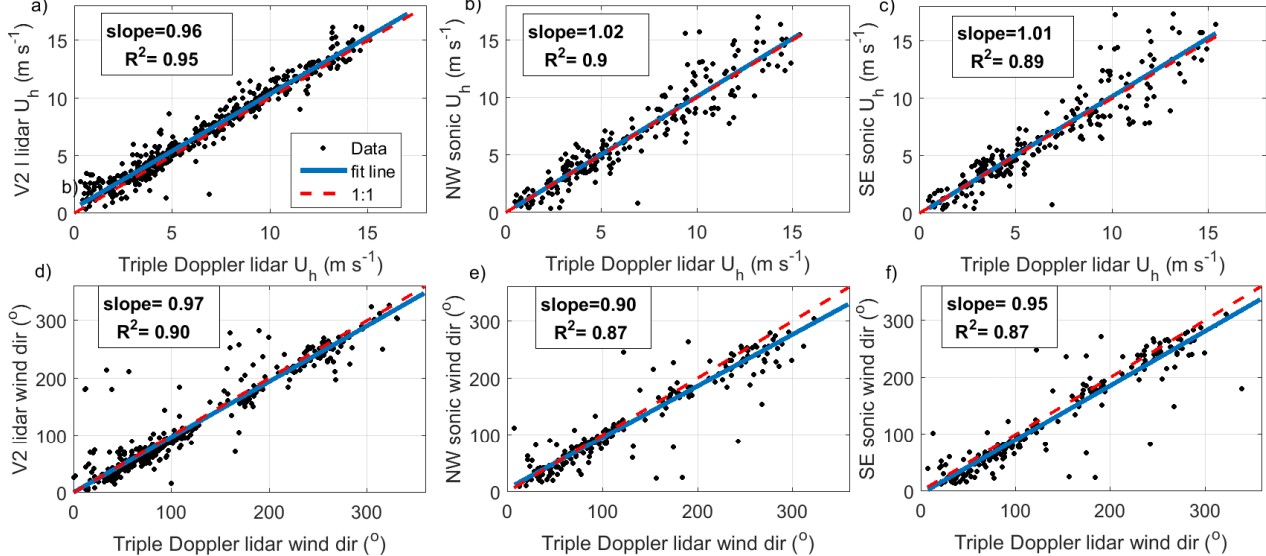

**Figure 8.** Linear regression of triple-Doppler lidar data against reference instruments for all the tested heights: a), b), c) Horizontal wind speed $U_h (m\ s^{-1})$; d), e), f) wind direction ($^o$).

Finally, histograms of the difference between the horizontal wind speed and direction measured through the triple-Doppler lidar measurements and the reference instruments are reported in Fig. 9. For the horizontal wind speed, the mean difference is of -0.38 m s$^{-1}$, -0.06 m s$^{-1}$, -0.09 m s$^{-1}$s and standard deviation of 0.83 m s$^{-1}$, 1.43 m s$^{-1}$, 1.60 m s$^{-1}$ with respect to V2 lidar, NW sonic and SE sonic, respectively. A similar analysis for the wind direction leads to a mean difference of 3.36$^o$, 7.47$^o$,
5    11.14$^o$ with standard deviation of 25.68$^o$, 26.09$^o$, 27.15 $^o$ compared with V2 lidar, NW sonic and SE sonic, respectively.

## 5   Comparison between lidar and radar virtual tower measurements

After discussing the assessment of the virtual tower measurements against the reference instruments, namely sonic anemometers installed over the BAO met-tower and a lidar profiler, a direct inter-comparison between Ka-band radar and wind lidar data is now presented.
10    According to the linear regression analysis presented in sections 3 and 4, a very good level of agreement for both radar and triple-Doppler lidar data was observed with reference instruments, as detailed in Tables 6 and 7. Generally the slope obtained for the correlation analysis was very close to 1 for both measurement techniques in the estimate of wind velocity (between 0.97 and 1.08 for radar data and between 0.95 and 1.02 for lidar data) and wind direction (between 0.84 and 0.87 for radar data and between 0.9 and 0.97 for lidar data). Correlation between the virtual tower measurements and data obtained from sonic anemometers and the V2 lidar is always larger than R$^2$>0.91 for the radar measurements and R$^2$>0.87 for the triple-Doppler





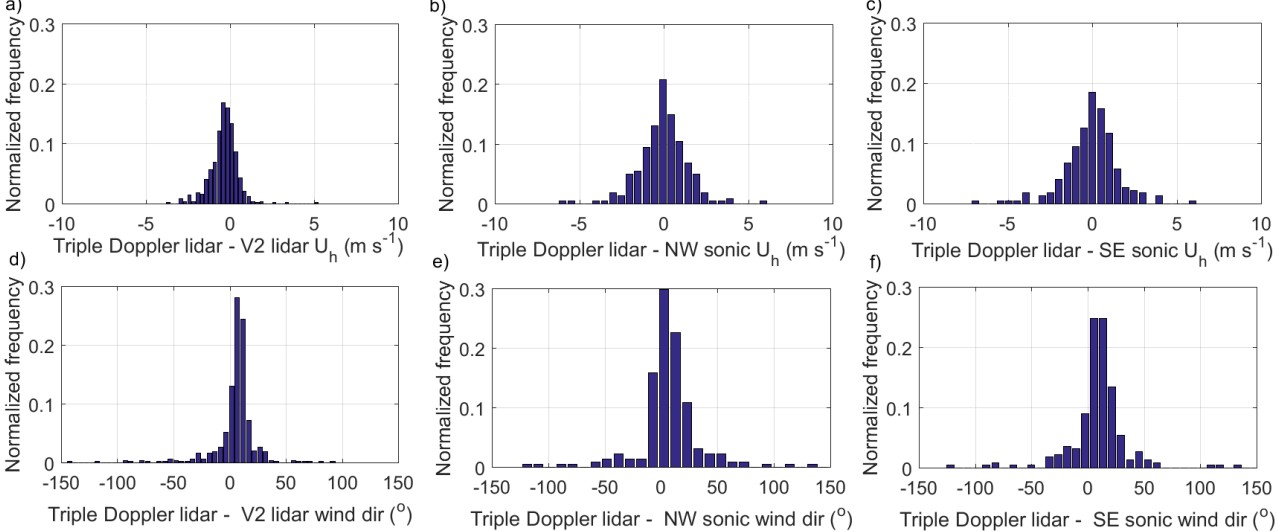

**Figure 9.** Difference of triple-Doppler lidar retrieval with reference instruments for all the tested heights: a), b), c) horizontal wind speed $U_h (m\ s^{-1})$; d), e), f) wind direction ($^o$).

lidar data. No systematic bias errors have been observed for both radar and triple-Doppler lidar measurements for the retrieval of the horizontal wind speed and wind direction (see Figs. 6 and 9).

In Fig. 10, a qualitative comparison between the wind data retrieved through the dual-Doppler radar measurements and the triple-Doppler lidar data is presented. Radar virtual tower measurements were performed continuously over the lidar supersite location with an average sampling period for each virtual tower of 3.3 s. Triple-Doppler lidar measurements, in contrast, were performed every 10 minutes due to a test schedule including other scans than these presented in this paper. A general good agreement is observed when virtual towers were performed simultaneously with the two Ka-band radars and the three scanning wind lidars. A similar variability in time and over the different heights was observed through the two different measurement techniques. Difference between the radar and the lidar measurements, for both horizontal wind speed and wind direction, is generally very small compared to the variability observed as functions of time and height.

In Fig. 11, statistics of the difference between the radar and lidar measurements are reported for the different virtual towers performed. For the wind velocity, the difference averaged over the height is always smaller than 0.5 m s$^{-1}$ with a maximum standard deviation of 0.29 m s$^{-1}$. For the wind direction, the maximum difference averaged over height is always smaller than 10$^o$, and the maximum standard deviation is 4.79$^o$.

# 6 Concluding remarks

During the XPIA experiment, co-located virtual tower measurements were performed with two Ka-band radars and three scanning Doppler wind lidars. Therefore, these tests provided the unique opportunity to perform a direct inter-comparison



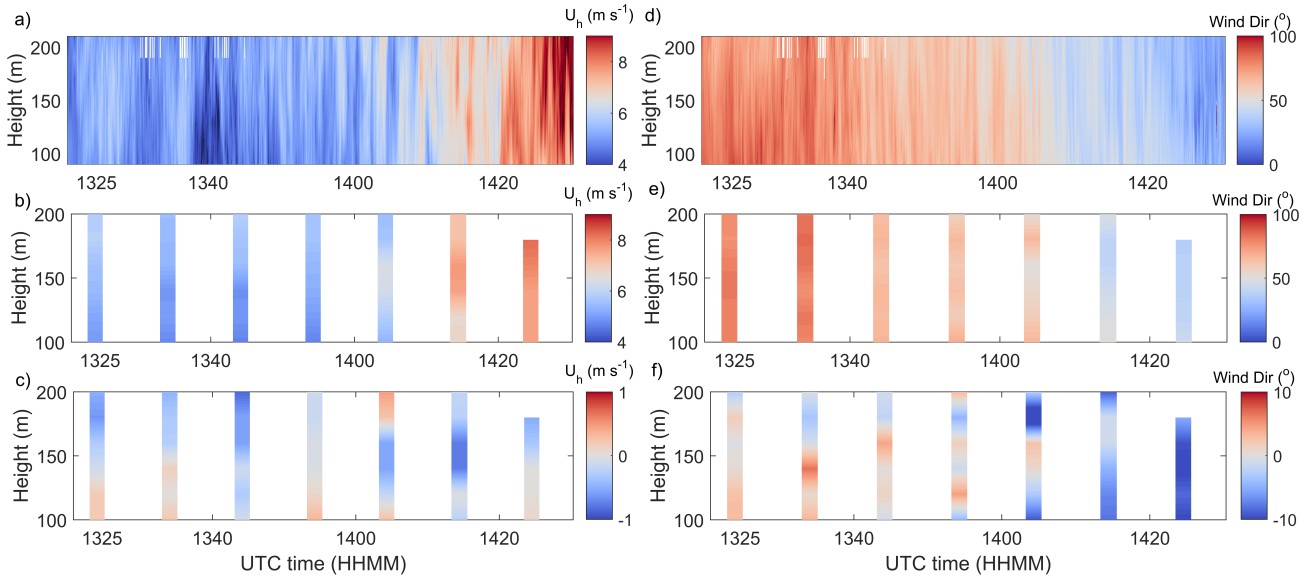

**Figure 10.** Inter-comparison between radar and lidar virtual tower measurements: a) horizontal wind speed $U_h\,(m\,s^{-1})$ retrieved with dual-Doppler radar; b) horizontal wind speed $U_h\,(m\,s^{-1})$ retrieved with triple-Doppler lidar; c) difference of horizontal wind speed $U_h\,(m\,s^{-1})$ between lidar and radar data; d) wind direction retrieved from dual-Doppler radar; e) wind direction retrieved from triple-Doppler lidar; f) difference in wind direction between lidar and radar data.

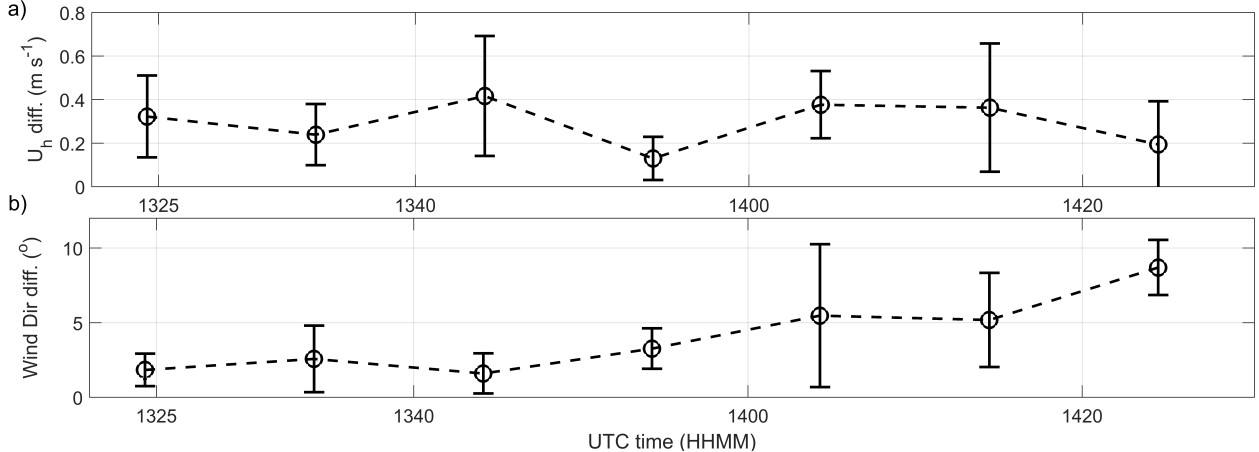

**Figure 11.** Statistics of the absolute value of the difference between radar and lidar wind data averaged over all the available heights: a) Horizontal wind speed $U_h\,(m\,s^{-1})$ difference; b) wind direction difference. Circles represent mean value, while error bars represent standard deviation.





between dual-Doppler radar and triple-Doppler lidar measurements. Furthermore, wind data obtained from the virtual tower measurements were also assessed against sonic anemometer data acquired from a met-tower located at a distance of 134 m from the virtual tower location and a lidar profiler that, in contrast, was co-located with the virtual towers.

Results of this assessment study show that, besides the use of different technologies, measurement volumes and sampling pe-
5 riods, multiple-Doppler radar and lidar measurements are both characterized by a good level of agreement with measurements performed with reference instruments, namely sonic anemometers and a lidar profiler. Through a linear regression analysis between virtual tower measurements, lidar profiler and sonic anemometer data, it was found that the slope is always within 0.84 and 1.02, while the correlation is always larger than $R^2$=0.87. No systematic bias errors have been detected for neither radar nor lidar measurements of the wind horizontal wind speed and direction. Regarding the vertical velocity retrieved through the
10 triple-Doppler lidar measurements, accuracy deteriorates rapidly with reducing height along the virtual tower, which is mainly a consequence of the lidar setup and the reduced elevation angles.

**Acknowledgments**

We express great appreciation to the numerous individuals and organizations who assisted with field deployments, including Erie High School, and the St. Vrain School District. We express appreciation to the NOAA/Earth System Research Labora-
15 tory/Physical Science Division for supporting the deployment of XPIA instrumentation at the BAO facility, and to the National Science Foundation for supporting the CABL deployments (https://www.eol.ucar.edu/field_projects/cabl). We would like to acknowledge operational, technical and scientific support provided by NCAR's Earth Observing Laboratory, sponsored by the National Science Foundation. Partial support for the UTD lidar and its deployment were provided by UTD institutional funds. NREL is a national laboratory of the U. S. Department of Energy, Office of Energy Efficiency and Renewable Energy, operated
by the Alliance for Sustainable Energy, LLC. Funding for this work was provided by the U. S. Department of Energy, Office of Energy Efficiency and Renewable Energy, Wind and Water Power Technologies Office, and by NOAA's Earth System Research Laboratory. The TTU measurement and analysis effort was provided through a contract with Sandia National Laboratories with funding from the US Department of Energy Wind and Water Power Technologies Office.





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
