# Peer review of "Assessment of virtual towers performed with scanning wind lidars and Ka-band radars during the XPIA experiment"

_Atmospheric Measurement Techniques, 2016_

## Referee Comment (RC1) · Anonymous Referee #1 · 13 Dec 2016

General comments

This manuscript presents an evaluation of virtual towers from Dual-Doppler-Radar and Triple-Doppler-Lidar measurements with ultrasonic anemometers and a Doppler lidar wind profiler. While those measuring techniques aren't new and other evaluations already exist (and cited by the authors), a comprehensive evaluation like this wasn't published yet to my knowledge. The comparison of virtual towers from triple-Doppler lidars and dual radars is novel, but sadly also very short and only qualitatively due to data availability. It would have been really interesting to have the same level of detailed analysis there as for the comparison to the reference instruments.

I have done Dual- and Triple Doppler lidar measurements and retrievals myself and I

didn't find major flaws in this work. The description of the measurement setup is mostly detailed enough for the purpose of the evaluation, but some information is missing (see specific comments). The authors use basic statistics for the evaluation and show the underlying data. The drawn conclusions agree to the data shown and can be followed.

With the disclaimer that I'm not native English speaking, I find the manuscript is well written. I noticed only a few minor mistakes.

Specific comments

Page 2, line 1: I think it would be a good addition here to explain briefly why Doppler lidars/radars only measure the line of sight velocity (or cite a work where it is explained).

Page 2, line 10: Maybe more precise "3D nature of the atmospheric boundary layer wind field"?

Page 2, line 14-16: The authors are clearly aware of this, but it should be written in the text that the accuracy of the retrieval is a function of not only the beam intersect geometry, but also the measurement errors of the individual instruments going into the retrieval.

Page 3, line 14: Does the 3 m leg spacing of the tower has any importance like less tower effects on the wind field?

Page 4, line 1-6: Please provide more information about this reference instrument (accuracy, serial number, year of production. . .).

Page 5, line 4: What was the pulse length of these two Doppler lidars?

Page 5, line 17 and 21: Accuracy or precision?

Page 5, table 3: What kind of errors (percentage, standard deviation) and do they have a unit? Also a sentence explaining that this is the error of a single lidar propagated through the rotation and being boosted up for small angles between the laser-beams could be considered here.

Page 6, line 3-4: If the sampling period at each point was 25 seconds and there are 6 levels between 100m and 200m, than I would expect the total time for a virtual tower to be longer than 127s?

Page 6, line 7: I think it would be important to include information how the north orientation of the lidars/radars was determined, as this is very important for correct beam intersect.

Page 8, line 6: I associate the term confidence level with results of tests of significance in statistics, which were not made here. The authors might think about rephrasing with other words.

Page 8, line 15-17: This sentence is unclear to me. I understood: The profiles of wind speed and direction where vertically interpolated to the heights of the ultrasonic anemometers. Is this correct?

Page 11, line 16: Is there any reason to assume it wouldn't be working for higher velocities or does this statement just refer to the velocity range during the measurements?

Technical comments

Page 1, line 18-19: Sentence structure.

Page 2, line 3: Maybe write out atmospheric boundary layer? The abbreviation is not used later on.

Page 3, line 7: "from" instead of "since"?

Page 3, line 10: Abbreviation CNR not introduced yet (it is written out on page 5, line 11 later).

Page 5, line 6: Missing brackets for citation.

Page 6, line 16: Missing brackets for citation.

Figure 5b: Lines of identity and linear regression are behind data points.

Page 11, line 18: Remove crisply.

Page 12, line 1: Delete first "wind" in "retrieval of wind horizontal wind speed".

Figure 8b: Inconsistent rounding compared to table 7.

Figure 10: Labels on the color map of direction are missing.

[Figure]

---

## Referee Comment (RC2) · Anonymous Referee #2 · 16 Dec 2016

General Comments:

The paper addresses an interesting evaluation of wind measurements with sonic anemometers, a profiling lidar, 3 scanning lidars and 2 Ka-band radars. The data records are generally very short, especially for the radar observations and it thus remains a bit unclear if the observed accuracy can also be reached under more variable conditions. I think the paper still merits publication but I would like the authors to provide more information about why only ~2hrs of data are available for the radars and also expand the discussion by addressing possible limitations under different types of conditions. It would also have been nice to see some recommendations in the conclusions or outlook sections about how different types of instruments can be best combined in

future studies for getting the most complete picture of boundary-layer processes. The authors only addressed the performance in terms of mean wind conditions while turbulent flow quantities are also very important characteristics. Could combining lidars with radars be helpful for also obtaining higher order velocity statistics?

Specific Comments:

p.3, 1. paragr. and p.4 l10-15: the authors mention that the measurements were performed over a period of ∼1w, but then later state that the radar data were actually only available for ∼2hrs. I find this quite misleading and the authors should be more specific here and also explain why the radars data were available only for such a short time window. What caused the radar data outside of this 2-hr window to be of poor quality and based on which criteria was it decided that the quality is acceptable/non-acceptable. Providing this information is critical as the reader otherwise does not know if the authors just picked a time period for which the agreement was best and excluded data for periods when the instruments were generally working but the agreement wasn't as good as expected. It would also be nice to add a short description of the general weather conditions for the day that was chosen for the comparison

p.4 the description of the instruments, the radars in particular, is rather short and possible limitations are not really discussed. What is the typical range of these instruments for different weather conditions, under which conditions do they not provide good data at all, etc.? Such information is very important and the authors should expand their description accordingly.

I would suggest combining Tables 1 and 2 into one table

p. 5, text below Eq. (1): it would be good if the authors can provide a reference for the error retrievals that are discussed here. Also, what was the reason for not having one of the scanning lidars continuously point vertically, which would have allowed to get much better observations for w?

[Figure]

p. 6 1. paragr.: what was the reason for the lidars not being synchronized and was there a minimum threshold for the overlapping time that was applied such that if the overlap time was below a certain value the data were ignored? Also, how high was the bias correction that was applied?

p. 8, l10-11: do the sonic data justify the assumption of a zero vertical velocity? I would suggest adding a panel with w-observations to some of the figures, such as e.g. Figure 7. It would also be of value for the discussion of the results in Table 7.

Figures 3 and 4: What are possible causes for the differences between the 2 sonics between 1320-1400UTC? The sonics do not seem to fall in the sectors where tower wakes could play a role during this time period.

p. 8, l 16: was the interpolation not applied to the data shown in Fig. 4? I would assume that the interpolation had to be first applied as the measurement heights would otherwise not match up but in the text it sounds like the interpolation was applied after these more qualitative comparisons.

p. 16, conclusions: given the short records of data the authors really must comment on how representative the observations and achieved accuracy are. Comments about possible challenges would also be helpful. The fact that only 2hrs of radar observations were found to be of acceptable quality lets me conclude that there may be quite a few challenges and for future studies and for deciding about the best strategies in obtaining boundary layer wind information a critical assessment of the pros and cons of different instruments is very important.

Technical Comments:

In several places (e.g. p. 5, l6-7) references are not placed correctly in parenthesis; the authors should carefully check and correct the references throughout the paper.

In the abstract, the word "performed" is used in several sentences and the authors should consider replacing it sometimes by a different word.
p.1, l9: how did the authors decide that the accuracy was great, what are the criteria that are used for concluding about the quality of the agreement?

p.2, l30: the article "the" is repeated twice

p.3, l7: use "from" instead of "since"

p. 3, l10: CNR must be defined, also would suggest using SNR instead. It is defined in l 11 on p.5 but should be defined when it is first used.

Captions of Figs 3 and 4: the authors should add information about the date of the observations

Figure 3: it would be nice if the authors can mark/highlight the time period for which the radar data are available

Table 5: were data from the sonics within the tower wakes included or excluded in the statistical analysis, please specify.

p. 8, l7-9: I am not quite sure what the authors try to say here, could this be simplified to something along the lines of: Given the good agreement between the sonic anemometers and the profiling lidars we felt confident that the data sets from these two types of instruments can be used to evaluate the accuracy of virtual tower measurements with scanning radars and lidars?

p. 10, l12: light of sight should be changed to line of sight

p. 11, l10: would suggest finding a better word than "prefigured" such as e.g. "expected"

---

## Author Comment (AC1) · 9 Feb 2017

We thank the Reviewer for his/her comments and the constructive review. Our replies are reported below.

1. *Page 2, line 1: I think it would be a good addition here to explain briefly why Doppler lidars/radars only measure the line of sight velocity (or cite a work where it is explained).*
   **Reply**: It is now added to the text: "A Doppler-based remote sensing instrument allows measurements of the wind velocity component parallel to the direction of the emitted wave source, e.g., a laser beam for a lidar or radio waves for a radar.

The measured wind velocity, which is referred to as radial or line-of-sight velocity, is proportional to the Doppler shift on the backscattered signal generated by the aerosol suspended in the atmosphere [Pena et al. 2013]". Reference added: Alfredo Pena, Charlotte B. Hasager, Julia Lange, Jan Anger, Merete Badger, Ferhat Bingol, Oliver Bischoff, Jean-Pierre Cariou, Fiona Dunne, Stefan Emeis, Michael Harris, Martin Hofsass, Ioanna Karagali, Jason Laks, Soren Larsen, Jakob Mann, Torben Mikkelsen, Lucy Y. Pao, Mark Pitter, Andreas Rettenmeier, Ameya Sathe, Fabio Scanzani, David Schlipf, Eric Simley, Chris Slinger, Rozenn Wagner and Ines Wurth, Remote Sensing for Wind Energy, DTU Wind Energy-E-Report-0029(EN), ISBN: 978-87-92896-41-4, 2013.

2. *Page 2, line 10: Maybe more precise "3D nature of the atmospheric boundary layer wind field"?*
   **Reply**: We mean a 3D variability in the ABL of the time-averaged wind velocity field, such as in presence of wind shear, veer, or wakes produced by upwind obstacles (e.g. wind turbines, buildings, topography), or for stratified wind turbulence (Segalini and Arnqvist 2015, JFM, 781, 330-352)

3. *Page 2, line 14-16: The authors are clearly aware of this, but it should be written in the text that the accuracy of the retrieval is a function of not only the beam intersect geometry, but also the measurement errors of the individual instruments going into the retrieval.*
   **Reply**: This comment has been added to the text.

4. *Page 3, line 14: Does the 3 m leg spacing of the tower has any importance like less tower effects on the wind field?*
   **Reply**: For the XPIA experiment, accuracy in the sonic anemometer measurements and possible tower effects have been discussed in the paper McCaffrey et al. 2017, now accepted for publication in the same AMT special issue of this manuscript. A reference to this paper is reported at page 3.

5. *Page 4, line 1-6: Please provide more information about this reference instrument (accuracy, serial number, year of production . . .).*
   **Reply**: the V2 lidar was a Windcube Offshore 8.66, unit WLS-16, with an absolute mean deviation smaller than 0.1 m s$^{-1}$ in wind speed and smaller than 2$^\circ$ in wind direction.

6. *Page 5, line 4: What was the pulse length of these two Doppler lidars?*
   **Reply**: The pulse length for the Doppler lidars is 200 ns. This detail is now added to the manuscript.

7. *Page 5, line 17 and 21: Accuracy or precision?*
   **Reply**: It is accuracy.

8. *Page 5, table 3: What kind of errors (percentage, standard deviation) and do they have a unit? Also a sentence explaining that this is the error of a single lidar propagated through the rotation and being boosted up for small angles between the laser-beams could be considered here.*
   **Reply**: As explained in the text (page 5, lines 17-22), error analysis is performed through the L2-norm of the rows of the matrix in Eq. 1. Error in the retrieved velocity components increases for values diverging from 1. The numbers in table 3 are dimensionless.

9. *Page 6, line 3-4: If the sampling period at each point was 25 seconds and there are 6 levels between 100m and 200m, than I would expect the total time for a virtual tower to be longer than 127s?*
   **Reply**: We apologize for this wrong information. The average time required for a virtual tower is 151.6 s.

10. *Page 6, line 7: I think it would be important to include information how the north orientation of the lidars/radars was determined, as this is very important for correct beam intersect.*

**Reply**: Estimate of the azimuthal bias from north for each lidar was retrieved through hard-target tests performed by hitting reference towers present on site with the lidar laser beam, and using their GPS coordinates with respect to the lidar location. These details are now added at the beginning of page 6.

11. *Page 8, line 6: I associate the term confidence level with results of tests of significance in statistics, which were not made here. The authors might think about rephrasing with other words.*
**Reply**: We absolutely agree with the Reviewer's comment. It is now reported: "Given the good agreement between the sonic anemometers and the profiling lidars, we felt confident that the data sets from these two types of instruments can be used to evaluate the accuracy of virtual tower measurements with scanning radars and lidars.".

12. *Page 8, line 15-17: This sentence is unclear to me. I understood: The profiles of wind speed and direction where vertically interpolated to the heights of the ultrasonic anemometers. Is this correct?*
**Reply**: The Reviewer is right.

13. *Page 11, line 16: Is there any reason to assume it wouldn't be working for higher velocities or does this statement just refer to the velocity range during the measurements?*
**Reply**: We want simply emphasize that the measurement systems observed significant variability in wind speed and direction during the experiment.

14. *Page 1, line 18-19: Sentence structure.*
**Reply**: This sentence has been rephrased.

15. *Page 2, line 3: Maybe write out atmospheric boundary layer? The abbreviation is not used later on.*
**Reply**: Added in the text.

16. *Page 3, line 7: "from" instead of "since"?*
    **Reply**: Added in the text.

17. *Page 3, line 10: Abbreviation CNR not introduced yet (it is written out on page 5, line 11 later).*
    **Reply**: Added in the text.

18. *Page 5, line 6: Missing brackets for citation*
    **Reply**: Edited.

19. *Page 6, line 16: Missing brackets for citation.*
    **Reply**: Edited.

20. *Figure 5b: Lines of identity and linear regression are behind data points.*
    **Reply**: Figure 5 is edited.

21. *Page 11, line 18: Remove crisply*
    **Reply**: Removed.

22. *Page 12, line 1: Delete first "wind" in "retrieval of wind horizontal wind speed"*
    **Reply**: Removed.

23. *Figure 8b: Inconsistent rounding compared to table 7*
    **Reply**: Table 7 (now Table 6) has been corrected.

24. *Figure 10: Labels on the color map of direction are missing.*
    **Reply**: A revised figure is now provided.

---

## Author Comment (AC2) · 10 Feb 2017

We thank the Reviewer for his/her comments and the constructive review. Our replies are reported below.

1. *The data records are generally very short, especially for the radar observations and it thus remains a bit unclear if the observed accuracy can also be reached under more variable conditions. I think the paper still merits publication but I would like the authors to provide more information about why only 2hrs of data are available for the radars and also expand the discussion by addressing possible limitations under different types of conditions.*

**Reply**: The Ka-band radars were only funded to be on site for approximately one month of the experiment (March 2015) and to collect 60 hours of dual-Doppler data. The latter objective was accomplished prior to the end of the month. Unfortunately, a mechanical issue with one of the radars terminated participation of the radars in the experiment. There were also multiple radar objectives during the XPIA project, including the collection of virtual towers, dual-Doppler horizontal sectors, and point-stares. Radar operators attempted data collection every day, but the time of year during which the experiment was held severely limited the amount of clear air data collected by the radars due to a low concentration of scatterers at Ka band. On that particular day (25 March, 2015), the carrier-to-noise ratio was too low for data collection prior to the onset of precipitation. Once the precipitation began, virtual tower data were collected for a total of 113 minutes. Of the 113 minutes, 107 minutes were available for dual-Doppler analysis after quality control. The scanning strategy was then switched to horizontal dual-Doppler sectors to target other objectives. 68 minutes of dual-Doppler sectors were also collected on this day before the precipitation ended and data quality degraded to unusable. A more thorough discussion of the limitations of the Ka-band radars is provided in the first paragraph on page 4. Some of the above details were also added to this paragraph. The text now reads: Two Texas Tech University Ka-band (8.6 mm wavelength) mobile Doppler radars (Hirth and Schroeder, 2013; Hirth et al., 2015; Gunter et al., 2015) were deployed during XPIA. These Ka-band radars were designed to operate in a variety of weather conditions, including precipitation and clear air. As for most radars, data quality and maximum range are typically greatest during periods of precipitation. In such environments, the maximum range of data can often exceed 20 km (depending on the employed scanning parameters for a given experiment). Data quality and maximum range tend to be reduced in clear air conditions, but the magnitude of the reduction is highly dependent upon the concentration of clear air scatters (e.g. dust, insects). Typical ranges in clear air can vary between 3 and 10 km. Late spring,

summer, and early fall typically provide the best clear air environments with biological scatterers being limited during the remaining portions of the year. The accuracy of the dual-Doppler virtual towers from radar data has been shown to be fairly consistent across different atmospheric conditions above approximately 50 m AGL (Gunter et al. 2015). Below this level, dual-Doppler wind speeds tended to be slightly overestimated in heavy precipitation (Gunter et al. 2015). During the XPIA experiment, the Ka-band radars were on site for 30 days. Atmospheric conditions allowed for quality dual-Doppler data collection on 17 days. Dataset lengths were largely dependent upon data quality and project objectives. During this time, the following radar scanning parameters were employed: pulse repetition frequency of 15 kHz, pulse width of 20 $\mu$m s and range resolution of 15 m. Radar1 (Radar2) was deployed 3.192 km (3.9 km) northwest (north) of the BAO tower (Fig. 1). Considering the 0.33° half-power beam width of the radars, these distances yielded an azimuthal resolution of 18 m (22.5 m) for Radar1 (Radar2) at the BAO tower location. Simultaneous RHI scans were performed by focusing both radars over the lidar supersite location by setting the radar azimuthal angles reported in Table 1, sampling rate equal to 5 Hz and sampling period of 3.3 s. For the 25 March 2015 dataset, virtual tower data were collected at the onset of precipitation and persisted for 113 minutes before switching scanning strategies to accomplish additional objectives. After quality control, analysis of the radar measurements, wind data from the two Ka-band radars for the period 25 March 1320 UTC to 1507 UTC and for heights ranging from 10 m to 490 m height with 20 m interval were available for this particular study.

2. *It would also have been nice to see some recommendations in the conclusions or outlook sections about how different types of instruments can be best combined in future studies for getting the most complete picture of boundary-layer processes.*
**Reply**: The following comments are now reported in the conclusions. This assessment study has shown that multiple-Doppler scans performed with either scanning lidars or radars allow achieving high accuracy in the retrieval of the wind speed and wind direction. The Ka-band radars generally provide continuous radial velocity measurements out to the maximum range when distributed meteorological targets (water droplets, ice crystals, etc.) are present. Overall, The Ka-band radar system is characterized by higher carrier-to-noise ratio under clear-air condition (low aerosol concentration) and light precipitations. A limitation of Doppler radars compared to lidars is the effect of beam spread at large ranges. Indeed, for the radars a divergence angle of $0.498°$ results in a beam spread of 17.1 m at 2-km range and 85.5 m at 10-km range. The scanning lidars, in contrast, have poor signal quality under precipitations and the carrier-to-noise ratio strongly depends on the concentration of aerosol suspended in the atmosphere. However, lidars might have a higher data availability under non-precipitation conditions and typical aerosol concentrations. The divergence angle of the lidars is practically negligible, leading to a constant spatial resolution throughout the measurement range. Regarding the scanning capabilities, the Ka-band radars have a maximum angular velocity in the scanning of $30°/s$, while for the lidars it is only $8°/s$.

3. *The authors only addressed the performance in terms of mean wind conditions while turbulent flow quantities are also very important characteristics. Could combining lidars with radars be helpful for also obtaining higher order velocity statistics?*
   **Reply**: During the XPIA experiment we limited our analysis to time-averaged parameters of the wind velocity field, mainly due to the lack of full synchronization between the various instruments and the variable overlapping time.

4. *p.3, 1. paragr. and p.4 l10-15: the authors mention that the measurements were performed over a period of 1w, but then later state that the radar data were actually only available for 2hrs. I find this quite misleading and the authors should*

*be more specific here and also explain why the radars data were available only for such a short time window. What caused the radar data outside of this 2-hr window to be of poor quality and based on which criteria was it decided that the quality is acceptable/non-acceptable. Providing this information is critical as the reader otherwise does not know if the authors just picked a time period for which the agreement was best and excluded data for periods when the instruments were generally working but the agreement wasn't as good as expected. It would also be nice to add a short description of the general weather conditions for the day that was chosen for the comparison*

**Reply**: Explanation of the length of the datasets was added to the paragraph introducing the Ka-band radars on page 4 (see response to comment 1).

5. *p.4 the description of the instruments, the radars in particular, is rather short and possible limitations are not really discussed. What is the typical range of these instruments for different weather conditions, under which conditions do they not provide good data at all, etc.? Such information is very important and the authors should expand their description accordingly. I would suggest combining Tables 1 and 2 into one table.*

   **Reply**: A more in-depth description of the Ka-band radars is offered in the paragraph on page 4. Further details regarding the typical ranges as well as variability with respect to different weather conditions is also included (See response to comment 1).

   For the measurements performed on March 25, 2015, the maximum and minimum range for scanning Doppler lidars varied from 300 m up to 3000 m, while the carrier-to-noise ratio was between -50 dB and 3 dB. Only measurements with carrier-to-noise ratio larger than -25 dB were selected for data post-processing considered in this paper.

   Table 1 and Table 2 are combined into one table.

[Figure]

6. *p. 5, text below Eq. (1): it would be good if the authors can provide a reference for the error retrievals that are discussed here. Also, what was the reason for not having one of the scanning lidars continuously point vertically, which would have allowed to get much better observations for w*
   **Reply**: This reference is now added to the text. Simley, E., Angelou, N., Mikkelsen, T., Sjöholm, M., Mann, J., and Pao, L. Y.: Characterization of wind velocities in the upstream induction zone of a wind turbine using scanning continuous-wave lidars, J. of Renewable and Sustainable Energy, 8, 013 301, 2016.

   An ideal setup consists of one lidar measuring directly the vertical velocity, while the other two lidars should have an azimuthal shift of $90°$ and zero elevation angle. A limitation of this scan strategy is represented by the impossibility of setting the lidars measuring the two horizontal velocity components with a zero elevation angle for measurement points located at different heights. Furthermore, it is noteworthy that the lidar pointing vertically cannot measure for distances lower than double the range gate of the lidar. Another advantageous configuration consists in deploying the three lidars with a $120°$ azimuthal shift and elevation close to $45°$. For real experiments, it is very tricky to achieve an optimal setup of the lidars due to possible site constraints; therefore, during XPIA we focused on evaluating accuracy of multiple-Doppler scans for sub-optimal configurations.

7. *p. 6 1. paragr.: what was the reason for the lidars not being synchronized and was there a minimum threshold for the overlapping time that was applied such that if the overlap time was below a certain value the data were ignored? Also, how high was the bias correction that was applied?*
   **Reply**: The lidars used for XPIA are commercial lidars operated with a GUI provided by the lidar manufacturer. This GUI does not allow the control of the lidars through a master computer. As reported in Fig. 2, the minimum overlapping time is 1 s. All the used bias errors are reported in Table 3.

8. *p. 8, l10-11: do the sonic data justify the assumption of a zero vertical velocity? I would suggest adding a panel with w-observations to some of the figures, such as e.g. Figure 7. It would also be of value for the discussion of the results in Table 7.*
   **Reply**: The time-average of the vertical velocity is typically very close to zero, but instantaneous vertical velocity can differ from zero and it is related to wind turbulence and atmospheric stability. A new panel has been added in Figure 7 for the vertical velocity.

9. *Figures 3 and 4: What are possible causes for the differences between the 2 sonics between 1320-1400UTC? The sonics do not seem to fall in the sectors where tower wakes could play a role during this time period.*
   **Reply**: A possible justification might be related to the statistical steadiness of the measurements, namely a too short sampling period for the occurred turbulence level.

10. *p. 8, l 16: was the interpolation not applied to the data shown in Fig. 4? I would assume that the interpolation had to be first applied as the measurement heights would otherwise not match up but in the text it sounds like the interpolation was applied after these more qualitative comparisons.*
    **Reply**: We have measurements from all the instruments at 150-m height, thus no interpolation is needed.

11. *p. 16, conclusions: given the short records of data the authors really must comment on how representative the observations and achieved accuracy are. Comments about possible challenges would also be helpful. The fact that only 2hrs of radar observations were found to be of acceptable quality lets me conclude that there may be quite a few challenges and for future studies and for deciding about the best strategies in obtaining boundary layer wind information a critical assessment of the pros and cons of different instruments is very important.*

**Reply**: Conclusion has been expanded including discussions for the data availability and accuracy, as reported in Comment 1. We also added the following discussion: Given the challenges associated with the collection of dual-Doppler radar data in non-precipitating environments, future experiments could incorporate both disdrometers and particulate monitors to better characterize clear air and precipitating environments most conducive to radar data collection. Data availability for all systems might also improve later in the calendar year when a greater concentration of scatterers is naturally present in the atmosphere.

12. *In several places (e.g. p. 5, l6-7) references are not placed correctly in parenthesis; the authors should carefully check and correct the references throughout the paper.*
   **Reply**: References are now corrected.

13. *In the abstract, the word "performed" is used in several sentences and the authors should consider replacing it sometimes by a different word*
   **Reply**: Other synonyms are now used.

14. *p.1, l9: how did the authors decide that the accuracy was great, what are the criteria that are used for concluding about the quality of the agreement?*
   **Reply**: The quality of the agreement among the different measurement techniques has been assessed through different methods, such as histograms of differences reported in Figures 6 and 9, linear regression analysis reported in Figures 5 and 8 and Tables 5, 6 and 7, or direct comparison such as in Figures 7, 10 and 11. For instance, from the linear regression analysis, slope close to 1 and R-square value about 0.9 are considered as quantification of great agreement between different measurement systems.

15. *p.2, l30: the article "the" is repeated twice*
   **Reply**: This typo has been corrected.

16. *p.3, l7: use "from" instead of "since"*
    **Reply**: 'since' is replaced with 'from'

17. *p. 3, l10: CNR must be defined, also would suggest using SNR instead. It is defined in l 11 on p.5 but should be defined when it is first used.*
    **Reply**: For the Windcube 200S, the carrier-to-noise ratio (CNR) is directly provided, which is in a way similar to the definition of signal-to-noise ratio.

18. *Captions of Figs 3 and 4: the authors should add information about the date of the observations*
    **Reply**: The date is now added.

19. *Figure 3: it would be nice if the authors can mark/highlight the time period for which the radar data are available*
    **Reply**: The time period of radar data availability is now marked in Figure 3 with vertical dashed lines.

20. *Table 5: were data from the sonics within the tower wakes included or excluded in the statistical analysis, please specify.*
    **Reply**: Data within the tower wake are included.

21. *p. 8, l7-9: I am not quite sure what the authors try to say here, could this be simplified to something along the lines of: Given the good agreement between the sonic anemometers and the profiling lidars we felt confident that the data sets from these two types of instruments can be used to evaluate the accuracy of virtual tower measurements with scanning radars and lidars?*
    **Reply**: That sentence has been modified according to this comment.

22. *p. 10, l12: light of sight should be changed to line of sight*
    **Reply**: We apologize for this typo.

23. *p. 11, l10: would suggest finding a better word than "prefigured" such as e.g. "expected"*
    **Reply**: The 'prefigured' word has been replaced with 'expected'.